# Soluble Urokinase Plasminogen Activator Receptor (SuPAR) Analysis for Diagnosis of Periprosthetic Joint Infection

**DOI:** 10.3390/antibiotics13020179

**Published:** 2024-02-12

**Authors:** Sebastian M. Klim, Jürgen Prattes, Florian Amerstorfer, Tobias Niedrist, Christoph Zurl, Martin Stradner, Barbara Dreo, Gunther Glehr, Andreas Leithner, Mathias Glehr, Patrick Reinbacher, Patrick Sadoghi, Georg Hauer

**Affiliations:** 1Department of Orthopaedics and Trauma, Medical University of Graz, 8036 Graz, Austria; sebastian.klim@medunigraz.at (S.M.K.); mathias@glehr.com (M.G.);; 2Department of Internal Medicine, Division of Infectious Diseases, Medical University of Graz, 8036 Graz, Austria; 3Clinical Institute of Medical and Chemical Laboratory Diagnostics, Medical University of Graz, 8036 Graz, Austria; 4Department of Internal Medicine, Division of Rheumatology and Immunology, Medical University of Graz, 8036 Graz, Austria; 5Department of Surgery, University Hospital Regensburg, 93053 Regensburg, Germany

**Keywords:** suPAR, joint infection, synovia, inflammation, diagnostic biomarker

## Abstract

Soluble urokinase plasminogen activator receptors (suPARs) are a biomarker for inflammatory diseases. This study aims to investigate its diagnostic properties regarding periprosthetic joint infections (PJI). This retrospective cohort study included adult patients who underwent joint puncture for suspected PJI. The presence of PJI was determined according to the criteria of the European Bone and Joint Infection Society (EBJIS). Laboratory study analyses included the determination of white blood cells (WBC) in whole blood, C-reactive protein (CRP) in blood plasma, and suPAR in both blood plasma and synovial fluid. Appropriate diagnostic cut-off values were identified utilizing Youden’s J, and their diagnostic performance was determined by calculating the positive (PPV) and negative predictive value (NPV) for each marker. Sixty-seven cases were included in the final analysis. Forty-three samples (64%) were identified as periprosthetic joint infection (PJI) and twenty-four specimen (36%) were PJI negative cases. The PPV and NPV were 0.80 and 0.70 for synovial suPAR, 0.86 and 0.55 for CRP, 0.84 and 0.31 for WBC and 1.00 and 0.31 for plasma suPAR. Synovial suPAR showed a solid diagnostic performance in this study and has the potential to be an alternative or complementary biomarker for PJI. Further investigations in larger patient collectives are indicated.

## 1. Introduction

Periprostethic joint infection (PJI) is the inflammation of a joint due to colonization with microbial pathogens in the wake of arthroblastic surgery. With a prevalence of at least 2%, PJIs are a frequent problem in orthopedic routine. As the number of joint replacement surgeries continues to rise globally, the incidence of PJIs is also on the ascent, posing substantial challenges in clinical management [1]. The possible consequences of PJI may go as far as revision surgery with joint exchange and systemic infection, leading to sepsis. Periprosthetic joint infections are not only associated with a higher mortality, they are also a meaningful burden for the public health system and costs. Also, distinguishing between aseptic failures and true infections is paramount, as misdiagnoses can lead to inappropriate interventions, prolonged patient suffering and increased healthcare costs [2,3]. Therefore, the fast and accurate identification of PJI is of great importance and a comprehensive understanding of the diagnostic methodologies becomes imperative considering the diverse clinical presentations and complexities associated with PJIs.

Laboratory biomarkers play a central role for the diagnosis of PJI, which has been defined by the European Bone and Joint Infection Society (EBIJS). The current definition of PJI does not rely heavily on laboratory investigation of human blood samples. PJI is mainly diagnosed through an analysis of the non-blood body fluid in the joint cavity, the synovial fluid. The focal diagnostic marker for PJI is the count of white blood cells (WBC) in a synovial sample. PJI is likely if a number of 1500 cells/µL is exceeded, while the diagnosis is confirmed at cell counts above 3000 cells/µL [4]. The diagnostic performance of synovial count is the highest of all criteria used by the current definition. Both the diagnostic sensitivity and the diagnostic specificity are approximately 90%. In contrast, microbological analysis yields more unreliable results and requires multiple samplings due to great non-disease related variations [4]. Other synovial biomarkers have been discussed lately, including the C-reactive protein (CRP), Calprotectin, and α-Defensin. All three provide good diagnostic performance but are poorly harmonized (Calprotectin, α-Defensin) and may be misleading in cases of non-infectious joint inflammation [5]. So far, no alternative biomarker has emerged which has the potential to replace synovial cell count as the golden standard for the diagnosis of PJI.

The urokinase plasminogen activator (uPA) and its cell-bound receptor, urokinase plasminogen activator receptor (uPAR), play pivotal roles in the proteolytic activation and degradation of plasminogen to plasmin within the pericellular matrix during tissue remodeling [6]. These regulatory mechanisms are crucial for various cellular processes, including cancer cell behavior and immune system functions. Notably, uPAR also exists in a soluble form, known as soluble urokinase plasminogen activator receptor (suPAR), which is found in the blood plasma, cerebrospinal fluid, and synovial fluid of both healthy individuals and patients with cancer or inflammatory diseases [7]. SuPAR is a glycoprotein that plays a pivotal role in diverse physiological processes, extending beyond its classical association with the urokinase plasminogen activator (uPA) system. Structurally, suPAR is a truncated form of the full-length uPAR, resulting from the cleavage of its extracellular domain. The suPAR molecule consists of three domains: D1, D2, and D3. Notably, the D1 domain contains the critical region responsible for binding uPA, facilitating its involvement in the proteolytic activation of plasminogen. This interaction between suPAR and uPA regulates pericellular proteolysis, influencing tissue remodeling processes [8,9]. In addition to its role as a plasminogen activator, suPAR has been implicated in the modulation of inflammatory responses in various diseases [10,11]. The stable plasma concentrations of suPAR are unaffected by circadian changes and fasting [12,13]. Recent publications have explored the utility of suPAR levels using enzyme-linked immunosorbent assay (ELISA) tests in the context of sepsis, cardiac disease, pneumonia, autoimmune disorders, and other conditions [14,15,16,17,18].

However, limited evidence exists regarding the diagnostic performance of blood suPAR in the specific context of PJI. So far, only a single study, conducted by an Italian research group, has provided preliminary insights into this area by investigating suPAR in blood samples from patients with PJI [19]. The authors observed distinctive patterns in suPAR levels among infected and non-infected individuals, indicating the value of suPAR as a potential biomarker for PJI diagnosis. However, given the exploratory nature of this investigation, the need for further validation and broader exploration of suPAR’s diagnostic capabilities in diverse patient populations is evident. In addition, no published data can be found regarding suPAR in synovial fluid. Therefore, it is crucial to comprehensively address the diagnostic properties of suPAR in both blood plasma and synovial fluid for the detection and management of these clinically significant infections. By addressing this research question, we aim to contribute to the existing knowledge and enhance diagnostic accuracy of PJI.

## 2. Results

Within a time period from January 2019 to October 2021, a total of 60 patients were included, of whom five had two and one had three joint punctures. The population consisted of 29 females and 31 males with a median age of 73 (64–78) and 74 (60–80) years, respectively. The synovial fluid samples were drawn either from the knee (*n* = 55; 82%), the hip (*n* = 11; 16%) or the shoulder (*n* = 1; 2%). PJI was identified in 43 (64%) samples, while 24 (36%) were PJI negative.

The results of laboratory analyses are presented in Table 1. Synovial cell count was performed in all cases while results for the other markers were missing for nine (WBC), nine (CRP), thirty (serum suPAR) and four (synovial suPAR) samples. As expected, the synovial cell count was significantly higher in the PIJ group than in PJI negative specimen (*p* < 0.001). The concentrations of CRP in blood plasma and of synovial suPAR were also significantly higher in PJI (*p* ≤ 0.002) while the results for WBC counted in whole blood and suPAR measured in blood plasma did not differ between both groups (*p* ≥ 0.132). Correlation analysis with Spearman’s Rho showed a weakly significant positive correlation of age and suPAR measured in serum (*p* = 0.046) in the PJI negativegroup (only nine cases). Otherwise, no significant correlations were found.

The ROC analysis was performed for all markers but synovial cell count, the curves are shown in Figure 1. The most appropriate diagnostic cut-offs are displayed in Table 2. The candidate cut-offs for plasma CRP and synovial fluid had a YI between 0.44 and 0.49 with a PPV ≥ 0.8 and a NPV between 0.44 and 0.70. The corresponding YIs (0.28 and 0.04) and NPVs (0.39 and 0.31) for WBC and serum suPAR were considerably lower. Specificity and PPV for serum suPAR seemed to be highest for serum suPAR, but only one sample of the PJI group had a result greater than the cut-off while 25 were false negatives.

## 3. Discussion

This study investigated the diagnostic properties of suPAR in both plasma and synovial fluid for periprosthetic joint infection (PJI). Our findings revealed that synovial, but not plasma, suPAR is highly correlated with the presence of PJI. Synovial suPAR demonstrated a comparable diagnostic performance to plasma CRP which, as we like to assume for our study population, reflects PJI-induced inflammation.

To our knowledge, only one study examined suPAR in the context of PJI so far by comparing serum suPAR to other inflammatory blood markers in samples from 80 patients of whom 45 had PJI [16]. The results of that investigation disagree in parts with ours: while CRP was significantly elevated in the PJI group and showed a very similar ROC curve to the one we present here, the blood levels of suPAR were also markedly increased in individuals with PJI. The latter is clearly contradictory to our results. The plasma suPAR concentrations presented here are not only higher overall compared to Galliera et al.’s work (6 vs. 4 ng/mL), but our cohort’s PJI group also showed a slight trend towards lower suPAR concentrations in blood plasma than the PJI-negative patients. There are two possible explanations for this discrepancy. First, the biological age seems to affect the concentration of circulating suPAR. A recent investigation from New Zealand explored the relationship between suPAR as an inflammation marker and accelerated aging processes in multiple organ systems, the central nervous system, and physical performance. The study analyzed a population-representative birth cohort of 843 patients, performing plasma suPAR measurements at ages 37 and 45, using the same assay as we did for the present study. Elevated suPAR levels were significantly associated with various negative effects on the investigated parameters, including accelerated aging from 26 to 45 years (average difference of 6.4 years comparing the top to the bottom quintile) and structural signs of older brain age in the central nervous system (both *p* < 0.001). Functional performance also exhibited a significant correlation with suPAR levels, with higher suPAR levels associated with poorer balance, reduced grip strength, and increased self-reported physical limitations (all *p* > 0.001). These associations remained significant even after adjusting for sex, smoking, CRP, and current health conditions. The study’s results suggest a link between suPAR and accelerated cognitive and physical aging processes, providing valuable insights for early detection of age-related pathologies and optimizing preventive measures [20]. Hence, as the concentration of circulating suPAR might be an expression of biological age, the observed disagreement with the study discussed above might lie within age differences of the study populations. Unfortunately, Galliera et al. did not report any demographic properties of their study population. We noticed a significant correlation with age in the control group, but the sample size in this subset was small. The second possible reason for the differing results could very well be an analytical one: while both investigations used the same assay and it is validated for both, blood serum and blood plasma samples, the manufacturer states an imprecision of 6% between different ELISA plates. In addition, the two investigations might have used different product charges of suPAR ELISA kits which would further contribute to differences of absolute laboratory results. However, since the studied sample size is smaller than in the other study (37 vs. 80 cases), the clinical relevance of suPAR measured in human blood samples during PJI remains doubtful and subject to further investigation.

Other studies have focused on exploring the potential of serum suPAR as a measurement tool for disease progression and joint destruction in rheumatoid arthritis (RA). Enocsson et al. recently demonstrated a significant correlation between baseline disease activity, joint damage at 36 months, and suPAR concentration in serum. However, suPAR levels did not exhibit predictive value. The authors concluded that close monitoring of patients with blood suPAR levels may be beneficial for detecting early stage joint destruction [21]. In general, circulating suPAR increases in the wake of systemic inflammation as it was shown in other inflammatory disorders like sepsis, cardiac disease, pneumonia, and autoimmune disorders [14,15,16,17,18].

Serum or plasma suPAR continues to be a focal point in clinical biomarker research. In June 2022, Holstein et al. published a study further investigating elevated suPAR levels in elderly patients in the emergency department and its predictive performance for 30-day mortality. The study suggests that suPAR could serve as a potential parameter to aid in decision-making regarding whether a patient requires inpatient care or can be safely discharged. The study included 1858 patients and used the same ELISA kit as our study. Elderly patients (>75 years) exhibited significantly higher suPAR values (5.4 ng/mL vs. 3.7 ng/mL, *p* < 0.001) compared to younger patients (<75 years). SuPAR correctly predicted all-cause 30-day mortality across all age groups. Consequently, different cutoff values were determined to support the decision-making process for discharging emergency department patients: a cutoff of 4 ng/mL was deemed safe for all patients. Taking the generally higher suPAR levels in the elderly subgroup into account, a cutoff of 5 ng/mL should be considered for this specific population [22]. This approach was also used during the COVID-19 pandemic when Stauning et al. were able to define a suPAR cut-off (<2.0 ng/mL), which could help the attending physician in the emergency department to separate expected mild from severe courses of disease, and thus, facilitate evidence-based triage [23]. Furthermore, suPAR is being used to evaluate new COVID-19 therapies, including a large, double-blind, randomized controlled phase 3 trial of anakinra, a recombinant IL-1 receptor antagonist. The research team established a cutoff of plasma suPAR ≥6 ng/mL for identifying patients at increased risk of respiratory failure progression. The authors concluded that early treatment with anakinra guided by plasma suPAR levels significantly reduced the risk of worse clinical outcomes on day 28 in patients with moderate and severe COVID-19 [24].

The novelty of our study lies in the measurement of suPAR in synovial fluid where we observed significantly elevated concentrations in samples drawn from patients suffering from PJI. Literatur research revealed only one original publication addressing suPAR in synovial fluid specimen: Chu et al. observed that synovial suPAR was significantly higher in samples from patients with gouty arthritis than in those with osteoarthritis [25]. The measured mean concentration of synovial suPAR in gouty arthritis in that paper (8.1 pg/µL) was markedly lower than the median synovial suPAR in the PJI samples in the present study (118 ng/mL). This could result either from analytical differences or might suggest that different causes of inflammation cause distinctive suPAR elevations. Gouty arthritis develops mainly through mechanical tissue destruction caused by uric acid crystals while PJI is a bacterial infection. Unfortunately, Chu et al. could not provide any results for serum suPAR. Apart from synovial fluid, several publications indicate that suPAR might be a laboratory marker which specifically identifies inflammation in localized body compartments or their fluids. In one study, salivary suPAR was not only clearly elevated in patients with peridontitis, it did also not correlate with suPAR measured in the corresponding serum samples [26]. Multiple investigations in the last 10 years revolved around suPAR in pleural fluids. Pleural suPAR is capable of distinguishing cardiac from non-cardiac pleural effusions and is associated with clinical outcome in pleural infection [27,28,29,30].

Protein-based or clinical chemistry markers are a popular diagnostic tool for investigation of body fluids and usually favored. Compared to cytological and microbiological analyses, a visual evaluation via microscopy is not needed and, thus, the probability of individual errors and the variation between different examinators does not factor into the laboratory result. In this regard, the suPAR assay bears the potential to provide an important diagnostic benefit since it does not require personnel trained and experienced in cytological analysis. Also, the suPAR assay is manufactured as an automated turbidimetric method from the same company which would allow for reporting results within one hour after sample arrival in the laboratory [31]. In addition, the costs per sample (approximately 9EUR ) are quite low. However, that particular automated assay has not been validated for other materials than lithium heparin blood plasma. Analytical standardization and harmonization is also easier for non-cytological assays. Of course, this does not mean that such methods are free of analytical errors. Not all immunoassays are traceable to a harmonized or internationally recommended standard material which restricts the comparability of their results to studies using the same assay from the same manufacturer. Modern CRP assays are a prime example for good analytical harmonization, their results are widely comparable [32]. In contrast, no such standards are (yet) available for suPAR. Conveniently, most of the studies which investigated suPAR and published in the last 10 years used the same products as we did in the present study.

We want to underline the following limitations of our study: The research is confined to a retrospective cohort study conducted exclusively a single center. Consequently, the generalizability of the findings to broader patient populations and diverse clinical settings may be limited. The study is based on a sample size of 60 patients, which may limit the statistical power and generalizability of the observed results. The exclusion of patients under guardianship or underage introduces a limitation, potentially restricting the extrapolation of the study’s findings to these specific demographic groups. Additionally, this study did not assess the diagnostic performance of synovial suPAR in comparison to other synovial biomarkers used in PJI diagnosis.

## 4. Materials and Methods

This work was a retrospective cohort study. We hypothesized that suPAR measured either in blood or synovial fluid samples can be utilized as diagnostic biomarker for the diagnosis of PJI. The protocol of this study was approved by the local institutional review board and all patients who were included gave their written informed consent. This investigation was conducted in compliance with the World Medical Association Declaration of Helsinki and the Austrian Data Protection Act [33].

All patients who underwent diagnostic joint puncture due to suspected PJI at the University Department of Orthopaedics and Traumatology Graz were eligible for participation. General patient data were prospectively collected from the hospital digital medical records and stored in a pseudonymized manner. Patients under guardianship or underage were excluded from the study. The assignment of diagnosis (PJI or PJI negative) was performed according to the previously described European Bone and Joint Infection Society (EBJIS) criteria by a blinded researcher prior to the study measurements [4].

### 4.1. Sample Collection and Laboratory Analyses

Synovial fluid was collected via joint aspiration under sterile conditions according to a standardized procedure [34]. Synovial fluid was drained into primary sample tubes containing lithium heparin but no separation gel (*VACUETTE LH Lithium Heparin*, Greiner Bio-One International GmbH, Kremsmünster, Austria). Blood sampling included whole blood anticoagulated with ethylenediaminetetraacetate (EDTA; *VACUETTE K3 EDTA*, Greiner Bio-One International GmbH) and lithium heparin (*VACUETTE LH Lithium Heparin* with separation gel, Greiner Bio-One International GmbH). The latter was centrifuged with 2300× *g* for 10 min at room temperature and the supernatant plasma was used for routine clinical chemistry analyses. Additional plasma was obtained by centrifugation of the EDTA blood samples after hematology analyses were completed.

Synovial cell counting was performed via optical microscopy while WBC in EDTA whole blood were quantified using an automated analyzer which utilizes fluorescence flow cytometry (*XN-1000*, Sysmex Austria GmbH, Vienna, Austria). The concentration of CRP was determined in lithium heparin plasma on an automated clinical chemistry analyzer featuring a turbidimetric immunoassay (*Tina-quant C-Reactive Protein IV*, cobas 8000, Roche Diagnostics GmbH, Mannheim, Germany) which is traceable to an international reference standard material (*ERM-DA474/IFCC*, Institute for Reference Materials and Measurements, Joint Research Centre, European Commission). The automated methods are part of high-throughput laboratory routine diagnostics and the analytical quality has been controlled accordingly. The imprecision for WBC count in whole blood was 2.1–2.9% (four identical analyzers are used in laboratory routine), while quality control for CRP showed a variation of <3%. After routine laboratory analyses were completed, residual synovial fluid and EDTA-Plasma were stored at −80 °C until conduction of the study analyses.

The concentrations of suPAR were measured in EDTA plasma and in synovial fluid using an enzyme-linked immunosorbent assay (*suPARnostic AUTO Flex ELISA*, ViroGates A/S, Birkeroed, Denmark). The assay uses recombinant suPAR as antigen which is pipetted into wells containing anti-suPAR antibiodies. No international standard material is available for suPAR. This method’s imprecision ranged between 2.3% and 6% and the limit of quantification was 0.4 ng/mL. As past studies indicated, the reference interval for suPAR in blood serum is clearly located below 5 ng/mL and showed a slight increase with age [35,36]. All samples were prepared and analyzed according to the instructions of the assays’ manufacturer. Since it was expected that the suPAR levels in synovial fluid exceeded the linear range of the assay (0.4–14.2 ng/mL), a preliminary investigation was conducted to determine the most suitable dilution. Based on the results, all samples were a priori diluted 1:5 with diluent provided by the manufacturer. For samples still exceeding the linear range, further dilution steps (up to 1:40) were performed. The final results were calculated by multiplying the values with the respective dilution factor.

### 4.2. Statistical Analysis

All statistical investigations were performed utilizing Excel 2016 (Microsoft Corporation, Redmond, WA, USA) and SPSS version 29 for Windows (IBM Corporation, Armonk, NY, USA). Distribution of metric variables was determined with the Shapiro–Wilk test. Group results were either reported as mean ± standard deviation (normal distribution) or as median along with the interquartile range (IQR). Also, depending on the distribution, the t-test for independent variables (normal distribution) or the Mann–Whitney-U test were used for identifying statistical differences at a level of significance of 0.05.

Diagnostic performance was objectified with receiver operator characteristics (ROC) analysis. The most suitable diagnostic cut-off for each marker (WBC, CRP and suPAR) was chosen according to the highest Youden’s J. The diagnostic sensitivity (true-positive rate, TPR) and specificity (true-negative rate; TNR) were calculated as well as the positive (PPV) and negative predictive values (NPV) for every marker’s selected cut-off.

## 5. Conclusions

Synovial suPAR showed a solid diagnostic performance in this study and has the potential to be an alternative or complementary biomarker for PJI. An eventual benefit over synovial cell count needs to be identified in a randomized controlled trial. Apart from diagnostics, synovial suPAR may be useful as a non-cell-based surrogate marker for upcoming investigations focusing on joint pathology and could also re-define the understanding of synovial inflammation. In this regard, further research using specific experimental models (e.g., cell cultures) are necessary.

## Figures and Tables

**Figure 1 antibiotics-13-00179-f001:**
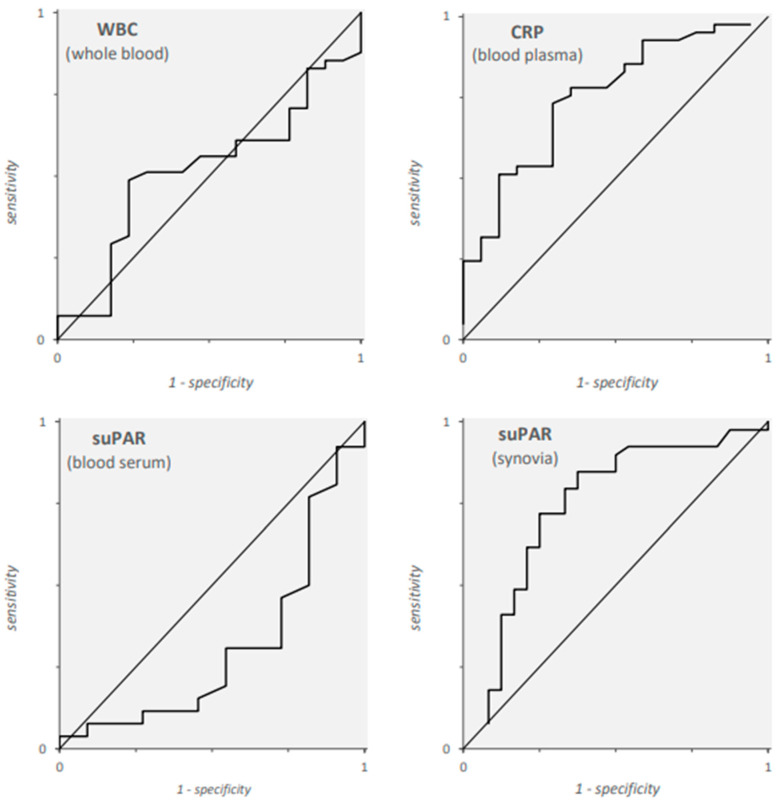
Receiver operator characteristics curves for white blood cells (WBC), C-reactive protein (CRP) and soluble urokinase plasminogen activator receptor (suPAR) regarding their ability to identify patients suffering from prosthetic joint infections.

**Table 1 antibiotics-13-00179-t001:** Laboratory results for patients with (PJI) and without prosthetic joint infection (aseptic). The results are reported as medians and the interquartile ranges *per* group. All laboratory values were rounded up. Statistical comparison was performed using the Mann–Whitney-U test, a *p*-value < 0.05 indicates a statistical difference between the groups. WBC = white blood cell. CRP = C-reactive protein. suPAR = soluble urokinase plasminogen activator receptor.

Analysis	PJI	PJI Negative	*p*-Value
blood			
WBC, G/L	10 (7–12)	9 (8–10)	0.713
CRP, mg/L	77 (38–152)	25 (10–70)	0.002
suPAR, ng/mL	5 (4–6)	8 (5–9)	0.132
synovial fluid			
WBC, cells/µL	25,000 (10,000–53,100)	250 (100–500)	<0.001
suPAR, ng/mL	118 (59–185)	38 (26–75)	<0.001

**Table 2 antibiotics-13-00179-t002:** Diagnostic performance characteristics of the measured markers regarding the identification of prosthetic joint infection. J = Youden’s J. TPR = true-positive rate = sensitivity. TNR = true-negative rate = specificity. PPV = positive predictive value. NPV = negative predictive value.

Analyte	Cut-Off	J	TPR	TNR	PPV	NPV
WBC (whole blood, *n* = 58)	9.6 G/L	0.28	0.51	0.76	0.84	0.39
CRP (blood plasma, *n* = 58)	36 mg/L	0.46	0.76	0.71	0.86	0.55
	75 mg/L	0.42	0.54	0.88	0.92	0.44
suPAR (blood plasma, *n* = 37)	12 ng/mL	0.04	0.04	1.00	1.00	0.31
suPAR (synovial fluid, *n* = 63)	54 ng/mL	0.49	0.82	0.67	0.80	0.70

## Data Availability

Study data available upon request.

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
