# Peer review of "Soluble Urokinase Plasminogen Activator Receptor (SuPAR) Analysis for Diagnosis of Periprosthetic Joint Infection"

_antibiotics, 2024, doi:10.3390/antibiotics13020179_

Round 1
Reviewer 1 Report
Comments and Suggestions for Authors
please find attached comments

minor correction in grammar and spelling is needed
Author Response
- This misunderstanding is caused by the difference of patients vs. samples à as mentioned in line 100-101, some patients had multiple joint aspirations.
- The term aseptic is changed to 'PJI negative' throughout the text for clarification.
- Unfortunately, we have no data on bacterial count for this study population.
- Grammar corrected, thank you.
- Corrected, thank you for noticing.
Reviewer 2 Report
Comments and Suggestions for Authors
I consider this manuscript suitable for publication but only after the authors address the following issues.
Table 2: why the first cut-off comes in G/L instead of g/L? Is it a different unit, not grams?
Throughout the text: put the “p” of “p-value” in italic and also the “vs” of “versus” and the “per” in italic. As conclusion the authors present only one small paragraph, they should extend this because of all the results and extensive discussion they present.
Minor issues:
- Line 21: do you mean “A blinded research”?
- Lines 37: PJI is or PJIs are.
- Line 40: do you mean “revision surgery”?
- Line 47: check the period at the end of the sentence.
- Line 48: “which is has been”, remove “is”.
- Line 107: to be consistent write “thirty” instead of “30”.
Comments on the Quality of English LanguageMentioned in the "Suggestions for Authors"
Author Response
- The unitage G/L is commonly used for quantitative results of blood leukocytes or platelets. The G stands for Giga and is a much more readable way of expressing 109 cells/L. In contrast, the prefered unit of blood erythrocytes count is T/L as in Tera (1012).
- The requested formatting was done throughout the whole draft.
- The paragraph containing the conclusion was extended.
- We corrected and/or rephrased the mentioned minor issues.
Thank you for reviewing our draft.
Reviewer 3 Report
Comments and Suggestions for Authors
The manuscript has written very well. I have minor comment as below that Author needs to address before publication.
1. Lines 27-28 grammatical error in Abstract. Forty-three (64%) were identified as periprosthetic joint infection (PJI) and 27 twenty-four (36%) were aseptic cases. Confusing statement please correct it.
Comments on the Quality of English Language
Minor editing required
Author Response
We changed the phrasing to "Forthy-three samples (64%) ..." and "... twenty-four specimen (36%) ..." for a better distinction of absolute and relative numbers.
Thank you for reviewing our draft.